# Human rights violations and associated factors of the Hijras in Bangladesh—A cross-sectional study

A. S. M. Amanullah[1], Tanvir Abir[2]*, Taha Husain[3], David Lim[4], Uchechukwu L. Osuagwu[5], Giasuddin Ahmed[6], Saleh Ahmed[7], Dewan Muhammad Nur -A Yazdani[8], Kingsley E. Agho[9]

1 Department of Sociology, Dhaka University, Dhaka, Bangladesh, 2 Department of Business Administration, Daffodil International University, Daffodil Smart City, Ashulia, Dhaka, Bangladesh, 3 Department of Gender and Development Studies, Begum Rokeya University, Rangpur, Bangladesh, 4 Health Services Management, School of Health Sciences, Western Sydney University, Sydney, Australia, 5 Translational Health Research Institute (THRI), School of Medicine, Western Sydney University, Campbelltown, NSW, Australia, 6 Dhaka Mass Rapid Transit Development Project (MRT Line- 1), Knowledge Management Consultants (KMC) Ltd, Banani, Dhaka, Bangladesh, 7 Bandhu Social Welfare Society, Dhaka, Bangladesh, 8 MONASH College at UCB, Dhaka, Bangladesh, 9 School of Health Sciences, Western Sydney University, Sydney, Australia

* t.abir73@gmail.com

**Data Availability Statement:** All relevant data are within the manuscript and its Supporting Information files.

## Abstract

### Background

Hijras in Bangladesh face considerable discrimination, stigma, and violence despite the 2013 legislation that recognized Hijras as a third gender. There is a dearth of published literature describing the extent of human rights violations among this population and their associated factors.

### Methods

A questionnaire was administered to 346 study participants aged 15 years and older, living in five urban cities of Bangladesh who self-identified as Hijra, in 2019. The six human rights violation indicators (Economic, Employment, Health, Education, Social and Civic and Political Right) assessed were categorized as binary. Associations between sociodemographic characteristics and the six human rights violations were tested using univariate and multivariate logistic regression.

### Results

Human right violations including economic, educational, political, employment, health and social/civil right violations were reported in 73.3%, 59.3%, 58.5%, 46.4%, 42.7%, and 34.4% of the participants, respectively. Economic rights violations were associated with bisexuality (Adjusted odds ratios [AOR] 3.60, 95%CI: 1.57, 8.26) and not living with family (AOR 2.71, 95%CI: 1.21, 6.09), while Hijras who earned more than 10,000 Bangladesh Taka experienced higher odds of educational (AOR 2.77, 95%CI: 1.06, 7.19) and political rights violations (AOR 4.30, 95%CI: 1.06, 7.44). Living in Dhaka city was associated with a reduced

**Funding:** NO: The authors did not receive support from any organization for the submitted work.

**Competing interests:** All authors certify that they have no affiliations with or involvement in any organization or entity with any financial interest or non-financial interest in the subject matter or materials discussed in this manuscript.

**Abbreviations:** AOR, Adjusted odds ratio; CI, Confidence interval; HIV, Human Immunodeficiency Virus; NGO, Non-governmental organization; OR, Odds ratio; SDG, Sustainable Development Goal; UDHR, Universal Declaration of Human Rights; UN, United Nation; WHO, World Health Organization.

odds for economic and political rights violation while experiencing violations of one human right could lead to violation of another in the Hijra community.

## Conclusion

Human rights violations were common in Bangladesh Hijras, particularly the Bisexual Hijras. Media and educational awareness campaigns are needed to address the underlying roots of a violation. Programs focused on the families, young people and high-income earners of this community are needed in Bangladesh.

## Introduction

Since the late 20th century, some activists and non-government organizations (NGOs) have lobbied for official recognition of the Hijras (refers to transgender, intersex, and effeminate homosexual people) as a kind of "third sex" or "third gender", being neither man nor woman [1, 2]. In a landmark achievement, the Bangladesh Government formally recognized the Hijras in 2013, making them eligible for priority education and low paid jobs [3]. However, this was cut short after the Ministry of Social Welfare tried to employ fourteen Hijras as office assistants in 2015, but still required the applicants to undertake a physical medical examination, where they concluded that the majority of the applicants who possess male anatomical genitalia were males and for one applicant without a penis, he was considered 'genetically male'. Following this, the Ministry terminated the appointments of all the Hijra applicants [4]. There are Hijras with penis and others without [5], and a common understanding is that Hijras are asexual, born with missing or ambiguous genitals, or "genitally handicapped" [4]. This misunderstanding may account for the Ministry of Social Welfare decision summarily dismissed the Hijra applicants in 2015.

The existence of Hijra is deeply rooted in Hinduism, including the deity Ardhanarisvara which is a composite male-female figure of Shiva and Parvati [6–8]. Gender variant individuals such as Amba/Shikhandin and Arjuna in the *Mahabharata* played a significant role in mythology. Historically, Hijras (then known as Khwaja Sara) were employed as custodians of the harem and held important positions in the court. The Hijras are central to Hindu practices, and as part of their badhai culture, they are often invited to the wedding, birth and other religious celebration [9]. The most important goddess for the Hijras is the Mother Goddess, Bahuchara Mata. In her name, Hijras perform their ritual function of giving blessings for fertility to a married couple or prosperity to a newborn child [4].

Bangladesh is a Sunni-majority country, and although historically a relatively tolerant and open-minded Muslim majority country, it remains conservative on homosexuals, bisexuals, and other gender and sexual diverse matters [10]. Although the Sunni *fatwa* (ruling on Islamic law) generally forbids gender reassignment surgery [11], the Bangladesh Government reached a landmark policy decision in 2013 that recognized Hijras as a third-gender, citing the universal human rights principles as a justification for the legislative change. The heteronormative concept of gender is dominant in Islam, and a trans person has to identify oneself as either a male or female [11]. The context is that each Hijra group in Bangladesh has both *janana* (non-emasculated) and *chibry* (emasculated) Hijra-members.

Despite comprising individuals of varying sex, gender, and sexuality, the Hijra community has often been referred to as 'female psyche in male physique' [12]. Past studies indicated that such stigma and discrimination drive social isolation, decrease economic support and lead to

poorer health and well-being [13]. The discrimination and the consequent vulnerabilities experienced by Hijras may lead to a higher risk of mental health problems such as long-term psychological complications from physical, verbal, and sexual abuse [14]. In practice, Hijras live an ostracized life in Bangladesh, work in working-class areas and have little interaction outside of their environment [4]. They find it challenging to find stable employment, housing, security, and social support. To survive, some are found begging for alms and engaged in the sex trade [15, 16] and face pervasive violence in public[17].

Whilst section 27 of *Bangladesh's Constitution* specifies that 'all people are equal before the law and entitled to equal legal protection', the fundamental enforcement of the recently recognized third gender's civil rights remains uncertain [18]. Some Hijras in Bangladesh are victims of rape, but unlike women and girls, their reports of rape are never filed because police officers do not believe that someone would harass this deviant group [19]. This highlights that Bangladesh's mainstream cultures cannot grasp and accept the multidimensional complexities of Hijras' diverse sex, gender, and sexuality [20]. The categorization of Hijras as a third-gender in Bangladesh and the legal recognition of innate genital difference as the marker of authenticity creates a false hierarchy over who is a real Hijra and further precipitates the marginalization [4].

Although there is increasing awareness of gender and sexually diverse communities in developing countries [13, 21, 22] and the Bangladesh society is working towards achieving a country where every person, irrespective of their gender and sexuality, can lead a quality life with dignity, human rights and social justice [23], there is a paucity of data on the human rights of Hijra communities[24]. Hence, this study explored the factors associated with six human rights violation indicators of Hijras in Bangladesh as well as provide baseline data for future evaluation of the policy effects through the recognition of Hijras as the third gender in Bangladesh. In addition, the findings will identify the subpopulation to target for future interventions to enable the Bangladeshi Government to meet the Sustainable Development Goals (SDGs 10) of reducing inequalities by a third through social inclusion among the Hijras by year 2030. The authors selected the components of human rights by following the major articles of *The Universal Declaration of Human Rights* (UDHR) proclaimed by the United Nations General Assembly in Paris on 10 December 1948 (General Assembly Resolution 217 A(III)). Civil, political, economic, cultural, health, and social rights are the legal bindings for the 192 member states of the United Nations [25].

## Methods

### Sample size determination

The required sample size was calculated using the statistical formula: $n_0 = z^2 p(1 - P)\big/_{d^2}$. Where $n_0$ is the sample size, $z^2$ was standard value normal distribution at 95% confidence level (1.96), proportion (p) = 0.50 was considered because previous studies among the Hijras conducted in Bangladesh are mostly qualitative, and $d^2$ was a maximum error (6%). Using an online sample size calculator (https://statulator.com/SampleSize/ss2PM.html [26]), the minimum sample size required was determined as 267, and considering a 30% non-response rate, a total sample size of 346 respondents will be required to detect statistical significance.

### Setting

This cross-sectional study was conducted between January 16–25, 2019, in five districts of Bangladesh (Dhaka/Gazipur, Chattogram, Mymensingh, Narayanganj and Rajshahi).

## Data collection

Prior to data collection, the enumerators underwent a 3-days training at the Bandhu Social Welfare Society premises, which was facilitated by the first author and two other experienced staff from Bandhu Social Welfare Society. The enumerators consisted of Hijras and people from other sexually diverse groups involved with the data collection. Data were obtained through face-to-face interviews from the Hijra at their House. The organization provided a list of 1600 Hijras from which a total 346 Hijras were randomly selected. Participants were selected from each of the five districts (Dhaka, Chattogram, Mymensingh, Narayanganj and Rajshahi) using probability proportional to size (PPS). PPS method was used due to the different district sizes. The Bandhu Social Welfare Society (BSWS) was founded two decades ago and it is responsible for addressing the health care needs and human rights issues of gender and sexual minority populations. BSWS aims to achieve a vision of a Bangladesh where every person, irrespective of gender and sexuality, can live a quality life with dignity, human rights and social justice [27].

## Study design

A structured questionnaire was developed, validated, and piloted prior to finalizing the study tool. The Kuder-Richardson (KR) Cronbach's alpha (α) for binary data was used to test the internal consistency of the survey tool and measure the reliability of the study. Table 1 shows the internal consistency level of the survey items, which indicates that the tool was appropriate for each build, with the α-value ranging from 0.70 to 0.88. The survey items included demographic characteristics of the participants and the six human rights violations of economic, employment, health, education, social/civil and political rights. Data were posted daily after fieldwork, which enabled daily review of work done to check for inconsistencies and errors.

## Outcome variables

The six human rights violations of economic, employment, health, education, political, social/civil rights were the outcome variables in this study. There were three items on the economic right violations, twelve on right to employment, six on right to health, five items each on violations to education and political rights, and eleven on social/civil rights of the participants (see Table 1 for details). For each of the items, a positive response was coded as '1' otherwise '0' for a negative response. A human right violation was recorded if the participant had responded positively to all of the items in that section (i.e. a coding of '1' to all items in that section). For example, on economic right, a violation was recorded only when 'the items Eco-1, Eco-2 and Eco-2 in Table 1 were each coded as'1', whereas any coding of '0' in Eco-1, Eco-2 or Eco-2 was considered no violation to the economic right of the respondent. Similar coding method was applied to the employment, health, education, political, social/civil right violations.

## Covariates

Table 2 shows the potential confounders in this study. They included the demographic characteristics of age, district (Dhaka and non-Dhaka), wages in Taka, Communities (Hijras and homosexuals), and educational status, working and living status. The six human rights violation variables were included when they were not used as outcome variables during the statistical analysis (Table 2).

## Ethics and consent

The study was approved by the Bandhu Social Welfare Society ethical committee. Data collected from the field was treated with high confidentiality, and prior to data collection, the

**Table 1. Variable identification, Kuder-Richardson (KR) cronbach's α of survey items and percentage responses of participants (n = 346).**

| Variables | Variable Definition | Positive response (%) | Cronbach's α |
|---|---|---|---|
| No permanent income | Eco-1 | 284 (82.1) | 0.74 |
| Did temporary jobs | Eco-2 | 288 (83.2) | 0.72 |
| No permanent job | Eco-3 | 286 (82.7) | 0.70 |
| **Economic right (all variables)** | Eco-All | | 0.79 |
| Resign jobs due to discrimination in the workplace | Emp-1 | 285 (82.4) | 0.82 |
| Education is a barrier to getting a private job | Emp-2 | 299 (86.4) | 0.82 |
| Education is a barrier to getting a public job | Emp-3 | 298 (86.1) | 0.82 |
| Dismissed from the job: Employer learnt of their feminine attitudes | Emp-4 | 289 (83.5) | 0.82 |
| Dismissed from the job: Employer thinks they are not suitable for the office environment | Emp-5 | 289 (83.5) | 0.82 |
| Victimized by other colleagues or officers | Emp-6 | 259 (74.9) | 0.83 |
| Employers think that they are not skilled enough | Emp-7 | 299 (86.4) | 0.83 |
| Employers think that they are not educated enough | Emp-8 | 284 (82.1) | 0.84 |
| Lack of coordination with employers | Emp-9 | 301 (87.0) | 0.84 |
| Lack of specific employment policies | Emp-10 | 311 (89.9) | 0.83 |
| The social taboo of the service recipients while getting services from them | Emp-11 | 317 (91.6) | 0.83 |
| Employers treated them as an unusual creature | Emp-12 | 331 (95.7) | 0.83 |
| **Employment Rights (all variables)** | Emp-All | | 0.84 |
| The presence of them is not tolerated by family members | CR-1 | 232 (67.1) | 0.86 |
| Scolded by the elders in the family/society | CR-2 | 243 (70.2) | 0.87 |
| Suffered verbal and mental abuse from society | CR-3 | 248 (71.7) | 0.87 |
| Forced to leave their family | CR-4 | 186 (53.8) | 0.86 |
| Faced isolation and rejection from society | CR-5 | 163 (47.1) | 0.86 |
| Faced denial of family property | CR-6 | 185 (53.5) | 0.86 |
| Parents, siblings and relatives were not comfortable to disclose their identity | CR-7 | 232 (67.1) | 0.86 |
| People showed a negative attitude and negligence to them | CR-8 | 323 (93.1) | 0.90 |
| Harassed by Law enforcing agencies | CR-9 | 287 (83.9) | 0.89 |
| Tough life due to being excluded from society | CR-10 | 321 (92.8) | 0.89 |
| Treated as if they are less than humans in society | CR-11 | 306 (88.4) | 0.89 |
| **Social and Civil Rights (all variables)** | CR-All | | 0.88 |
| Faced stigma and discrimination from the students | RE-1 | 298 (86.1) | 0.85 |
| Faced stigma and discrimination from the teachers | RE-2 | 235 (67.9) | 0.86 |
| Classmates scared and hated interacting and playing with Hijra Community | RE-3 | 275 (79.5) | 0.83 |
| Guardians of students not accepting coeducation of Hijra Community with their child | RE-4 | 262 (75.7) | 0.84 |
| Lack of enabling environment in the educational institutions | RE-5 | 290 (83.8) | 0.83 |
| **Right to education (all variables)** | RE-All | | 0.87 |
| Sexual and reproductive health is not adequately addressed | HR-1 | 305 (88.2) | 0.70 |
| No access to public health service | HR-2 | 254 (73.4) | 0.69 |
| Access to private health service | HR-3 | 200 (57.8) | 0.71 |
| Lack of education on the health concern among HCPs | HR-4 | 284 (82.1) | 0.70 |
| Unacceptance of women and sexual abuse of men in the hospital wards | HR-5 | 276 (79.8) | 0.71 |
| Faced mental and sexual harassment during treatment. | HR-6 | 310 (89.6) | 0.74 |
| **Health Rights (all variables)** | HR-All | | 0.74 |
| Authority does not emphasis on political participation. | PR-1 | 305 (88.2) | 0.60 |
| The rights specific to them is not included in the election manifesto | PR-2 | 299 (86.4) | 0.59 |
| Not included as candidates in elections and campaigns | PR-3 | 282 (81.5) | 0.63 |
| No separate voter list | PR-4 | 276 (79.8) | 0.76 |
| Face obstacles for obtaining NID and international passport | PR-5 | 285 (82.4) | 0.63 |
| **Political Rights (all variables)** | PR-All | | 0.70 |

**Table 2. Participant Characteristics and the categories of human right violations (n = 346).** The non-Dhaka district includes: Chattogram, Mymensingh, Narayanganj and Rajshahi districts.

| Variables | Frequency (n) | Per cent (%) |
|---|---|---|
| **Demography** | | |
| *Age in years* | | |
| 15–25 | 194 | 55.6 |
| 26+ | 155 | 44.4 |
| *District* | | |
| Non-Dhaka | 235 | 68.1 |
| Dhaka/Gazipur | 110 | 31.9 |
| *Hijra Communities* | | |
| Bisexuality | 200 | 57.8 |
| Homosexuality | 146 | 42.2 |
| *Educational status* | | |
| Primary | 70 | 20.2 |
| Secondary | 196 | 56.7 |
| Tertiary | 80 | 23.1 |
| *Working status* | | |
| Unemployed | 72 | 20.6 |
| Employed | 277 | 79.4 |
| *Wages in Taka* | | |
| 0–5000 | 73 | 21.3 |
| 5000–10000 | 164 | 47.8 |
| 10000+ | 106 | 30.9 |
| *Living status* | | |
| With family | 180 | 52.0 |
| Without family | 166 | 48.0 |
| **Human rights** | | |
| *Economic right* | | |
| No | 93 | 26.7 |
| Yes | 256 | 73.4 |
| *Health right* | | |
| No | 200 | 57.3 |
| Yes | 149 | 42.7 |
| *Employment right* | | |
| No | 187 | 53.6 |
| Yes | 162 | 46.4 |
| *Right to education* | | |
| No | 142 | 40.7 |
| Yes | 207 | 59.3 |
| *Social and Civil right* | | |
| No | 229 | 65.6 |
| Yes | 120 | 34.4 |
| *Political right* | | |
| No | 145 | 41.6 |
| Yes | 204 | 58.5 |

participants were informed of the confidentiality of the information they provide. Verbal and written consent was obtained from all the participants, and the participants' confidentiality and privacy were maintained. The study adhered to the tenets of Helsinki's Declaration and data collection tools focused on the UDHR values.

## Statistical analysis

Frequency tabulations were used to describe the potential covariates, and this was followed by the prevalence and 95% confidence intervals (CIs) of the six human rights violations in this study. Simple, bivariate and multiple logistic regression were performed to determine the factors associated with the six human rights violation outcomes. To assess the factors associated with each of the six human right variables, all the sociodemographic variables including the human right variables that were not used as the outcome variable in that regression, were taken as covariates. Thus, six logistic regression analysis corresponding to the six human right violation outcomes were conducted and their association with the demographic variables were tested using odds ratios (OR) and their 95%CIs. All analyses were carried out using STATA/MP version 14 (Stata Corp 2015, College Station, TX, USA) and $p < 0.05$ was considered statistically significant.

## Results

### Characteristics of the study population

Three hundred and forty-six Bangladeshi Hijras participated in this study, and the breakdown of their characteristics, including the proportion that reported human rights violations, are shown in Table 2. More than half of the respondents were aged 15–25 years, self-reported as being bisexual, had at least a secondary education and lived with their families. About 68% of them lived outside of the capital city Dhaka. Although the majority (79.4%) were employed, less than one-third earned more than 10,000 Taka at the time of this study, and about two-thirds reported violations of their human rights (73.4%).

### Prevalence of human right violations among the Hijra communities in Bangladesh

Fig 1 shows the prevalence and 95% CI of the six human rights violations reported among the Hijras in Bangladesh. About two-third (73%) of the participants had experienced economic rights violations, with more than half of them reporting violations to their political and educational rights. The prevalence of health and Social/civil rights violations was comparatively lower than other human rights violations among the Hijras.

### Univariate analysis of factors associated with the six human rights violations among the Hijras communities in Bangladesh

Table 3 shows the odds ratios and 95%CI of the factors related to violation of Human rights in the Bangladesh Hijras. Hijras who lived in the Dhaka region, those who completed university education, those employed and Hijras who earned more than 10,000 Bangladesh Taka at the time of this study reported lower odds of violating their basic human rights. Age was associated with human rights violations in this community with lower odds among those aged 26 years and over compared with younger people (<26 years). Hijras whom self-reported to be bisexual had lower odds of violations of their political rights.

There were significant associations between the different categories of human rights violations assessed in this study. The violation of one human right predicted the violation of

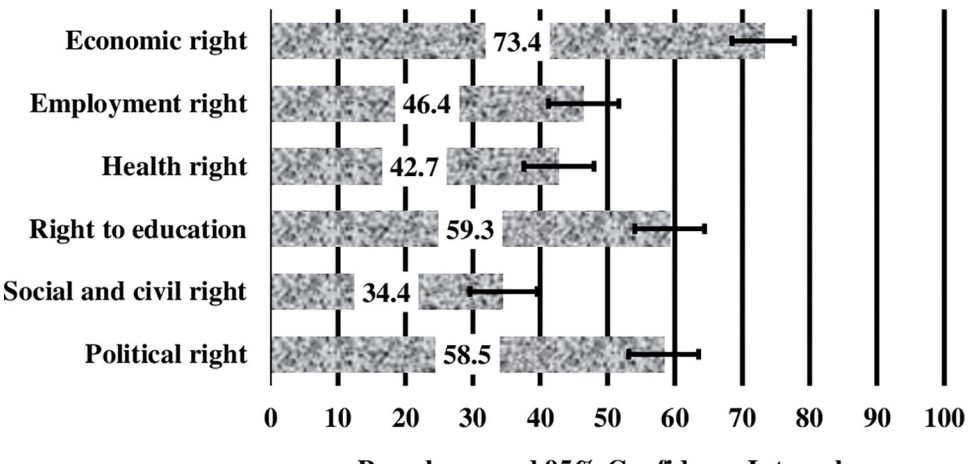

**Fig 1. Prevalence of six human rights violations of Bangladeshi Hijras.** Error bars represent 95% confidence intervals.

another human right category. For example, Hijras, who reported violations of their economic rights, had higher odds of violating their political rights (OR 3.22, 95%CI 1.97, 5.27). Those who reported violations to their right to good health showed higher odds of violations to their education (OR 4.21, 95%CI 2.62, 6.78) and Social/Civil rights (OR 2.39, 95%CI 1.52, 3.75).

## Multivariate analysis of six human right violations among the Hijras communities in Bangladesh

Table 4 presents the adjusted odds ratios and 95%CI of the factors associated with human rights violations in the Hijras. After adjusting for all the potential confounders, the Hijra community was more likely to experience economic rights violations but less likely to violate their political rights in this study. Hijras aged 26 years and over had lower odds of violations of their right to basic health than those younger than 26 years. Living without one's family and being bisexual was associated with a 2.71 (95%CI 1.21, 6.09) and 3.60 (95%CI 1.57, 8.26) increase in the odds of violation of economic rights in this community.

The odds for educational and political right violations were increased by 3 times (95%CI 1.06, 7.19) and 4 times (95%CI 1.06, 7.44) respectively, among Hijras who earned more than 10,000 Taka compared to those who made less than 5,000 Taka, after adjusting for the demographic factors. Participants who reported violations of their right to basic health had a higher likelihood of violations in other areas of their human rights (educational and social/civil rights). The odds of violation of the economic rights of the Hijras was increased by 4.6 times (95%CI 2.67, 7.91) compared with those who reported violations of their educational rights.

## Discussion

This study used cross-sectional data to identify the several human rights violations experienced by the Hijra community in Bangladesh and examined the factors associated with human rights violations. The study found that human rights violations were commonly reported by the Bangladesh Hijra community, with two-thirds reporting that their economic rights were violated and more than half had experienced violations of their political and educational rights. Economic rights violations were more likely to be reported by bisexual Hijras and those who were not living with their families, while Hijras who earned more than 10,000 Bangladesh Taka

**Table 3. Simple logistic analysis of six human rights violations in the Bangladeshi Hijras.** Odds ratios (OR) and 95% Confidence Intervals (CI). Confidence Intervals (CI) that exclude 1.00 are statistically significant at p<0.05 level and were bolded. Empty cells represent the human rights category that wasused as anoutcome variable for that model. The non-Dhaka districts included:Chattogram, Mymensingh, Narayanganj and Rajshahi districts.

| Variables | Economic rights | Employment rights | Health rights | Education rights | Social and Civil rights | Political rights |
|---|---|---|---|---|---|---|
| **Demography** | OR[95%CI] | OR[95%CI] | OR[95%CI] | OR[95%CI] | OR[95%CI] | OR[95%CI] |
| *Age in years* | | | | | | |
| 26+ (15–25, OR = 1) | 1.02 [0.63, 1.64] | 1.05 [0.69,1.60] | **0.56 [0.36, 0.86]** | **0.59 [0.38, 0.91]** | **0.55 [0.35, 0.87]** | 1.02 [0.66, 1.56] |
| *District* | | | | | | |
| Dhaka/ Gazipur (Non-Dhaka, OR = 1) | **0.15 [0.09, 0.26]** | 1.26 [0.80, 1.99] | 1.08 [0.69, 1.71] | 1.58 [0.98, 2.53] | 1.39 [0.87, 2.23] | **0.06 [0.03, 0.10]** |
| *Hijra Communities* | | | | | | |
| bisexuality (homosexuality, OR = 1) | 1.37 [0.83, 2.24] | .94 [0.61, 1.44] | 1.80 [1.17, 2.77] | 0.89 [0.58, 1.38] | 0.83 [0.53, 1.30] | **0.26 [0.16, 0.40]** |
| *Educational status* | | | | | | |
| Secondary (Primary, OR = 1) | 0.92 [0.47, 1.77] | 0.89 [0.52, 1.53] | 1.10 [0.63, 1.92] | 1.22 [0.71, 2.12] | 0.83 [0.47, 1.48] | 1.32 [0.76, 2.29] |
| Tertiary | **0.45 [0.22, 0.94]** | 0.60 [0.31, 1.14] | 1.36 [0.71, 2.60] | 1.65 [0.85, 3.20] | 1.26 [0.65, 2.45] | 1.49 [0.77, 2.85] |
| *Working status* | | | | | | |
| Employed (Unemployed, OR = 1) | 1.07 [0.60, 1.92] | 1.47 [0.87, 2.50] | 0.98 [0.58, 1.66] | **0.53 [0.30, 0.93]** | 0.78 [0.46, 1.34] | 0.70 [0.41, 1.20] |
| *Wages in Taka* | | | | | | |
| 5000–10000 (< 5000, OR = 1) | 1.91 [0.99, 3.70] | 1.31 [0.75, 2.29] | 1.19 [0.68, 2.08] | 0.97 [0.55, 1.69] | 0.84 [0.47, 1.50] | 1.10 [0.62, 1.96] |
| 10000+ | 0.58 [0.30, 1.10] | 0.95 [0.52, 1.73] | 1.21 [0.66, 2.21] | 1.38 [0.75, 2.54] | 1.05 [0.57, 1.96] | **0.44 [0.24, 0.81]** |
| *Living status* | | | | | | |
| Without family (With family, OR = 1) | 1.01 [0.62, 1.63] | 1.11 [0.73, 1.70] | 0.67 [0.44, 1.03] | 0.94 [0.61, 1.44] | 1.08 [0.69, 1.68] | 0.96 [0.62, 1.47] |
| **Categories of Human rights** | | | | | | |
| *Economic rights* | | | | | | |
| Yes (No, OR = 1) | - | 1.21 [0.75, 1.95] | 1.04 [0.65, 1.69] | 1.01 [0.62, 1.64] | 0.88 [0.53, 1.44] | **3.22 [1.97, 5.27]** |
| *Health rights* | | | | | | |
| Yes (No, OR = 1) | 1.04 [0.65, 1.69] | 0.75 [0.49, 1.15] | - | **4.21 [2.62, 6.78]** | **2.39 [1.52, 3.75]** | **0.65 [0.42, 0.99]** |
| *Employment rights* | | | | | | |
| Yes (No, OR = 1) | 1.21 [0.75, 1.95] | - | 0.75 [0.49, 1.15] | 0.91 [0.59, 1.39] | 0.92 [0.59, 1.43] | **0.57 [0.37, 0.88]** |
| *Right to education* | | | | | | |
| Yes (No, OR = 1) | 1.01 [0.62, 1.64] | 0.91 [0.59, 1.39] | **4.21 [2.62, 6.78]** | - | **3.48 [2.11, 5.73]** | 1.22 [0.79, 1.87] |
| *Social and Civil rights* | | | | | | |
| Yes (No, OR = 1) | 0.88 [0.53, 1.44] | 0.92 [0.59, 1.43] | **2.39 [1.52, 3.75]** | **3.47 [2.11, 5.73]** | - | 1.36 [0.87, 2.15] |
| *Political rights* | | | | | | |
| Yes (No, OR = 1) | **3.22 [1.97, 5.27]** | **0.57 [0.37, 0.88]** | **0.65 [0.42, 0.99]** | 1.22 [0.79, 1.87] | 1.36 [0.87, 2.15] | - |

were more likely to report violations to their educational and political rights. Those who experienced violations of one human right were more likely to report violations of another. The persistent violations of the Hijras human rights found in this study highlights the difficulties faced by the Bangladesh Government in meeting the UN SDGs 1, 4, 5, 8 and 10 of adequate access to education, inequality, poverty reduction, gender equality and economic growth among the Hijras people. Since Bangladesh is a signatory to the UDHR, the Government has an obligation to reduce the structural and cultural barriers to education for all its citizens.

Despite the Bangladeshi Government's landmark policy decision in 2013 to recognize the rights of the Hijras, less than half of the Hijras in this study enjoyed basic rights to health, economic and social/civil rights. This finding is consistent with a previous study in Bangladesh on social exclusion of the Hijras [12], which showed that social and economic exclusion of the Hijras resulted in negative health, limited or no access to social, educational, legal and health services [12]. The finding that Hijras who earned high wages were more likely to report violations to their educational rights was also consistent with a recent qualitative study among

**Table 4. Multiple logistic analyses of six human right violations in Bangladesh Hijras.** Adjusted odds ratios (AOR) and 95% Confidence intervals (CI). Confidence Intervals (CI) that exclude 1.00 are statistically significant at p<0.05 level and were bolded. Empty cells represent the human rights category that wasused as anoutcome variable for that model. The non-Dhaka districts included:Chattogram, Mymensingh, Narayanganj and Rajshahi districts.

| Variables | Economic rights | Employment rights | Health rights | Education rights | Social and Civil rights | Political rights |
|---|---|---|---|---|---|---|
| Demography | AOR[95%CI] | AOR[95%CI] | AOR[95%CI] | AOR[95%CI] | AOR[95%CI] | AOR[95%CI] |
| *Age in years* | | | | | | |
| 26+ (15–25, OR = 1) | 1.03 [0.56, 1.88] | 1.00 [0.62, 1.61] | **0.56 [0.34, 0.95]** | 0.84 [0.50, 1.41] | 0.63 [0.37, 1.06] | 0.81 [0.38, 1.70] |
| *District* | | | | | | |
| Dhaka/ Gazipur (Non-Dhaka, OR = 1) | **0.15 [0.06, 0.35]** | 1.13 [0.54, 2.39] | 0.66 [0.29, 1.52] | 1.84 [0.80, 4.20] | 2.06 [0.92, 4.62] | **0.01 [0.00, 0.02]** |
| *Hijra Communities* | | | | | | |
| bisexuality (homosexuality, OR = 1) | **3.60 [1.57, 8.26]** | 0.69 [0.37, 1.29] | 1.38 [0.71, 2.70] | 0.78 [0.39, 1.55] | 1.04 [0.54, 2.03] | **0.01[0.00, 0.04]** |
| *Educational status* | | | | | | |
| Secondary (Primary, OR = 1) | 0.53 [0.24, 1.16] | 1.02 [0.56, 1.85] | 0.89 [0.46, 1.72] | 1.37 [0.72, 2.62] | 0.86 [0.45, 1.65] | 1.13[0.43, 2.95] |
| Tertiary | **0.22 [0.08, 0.57]** | 0.85 [0.40, 1.81] | 0.95 [0.42, 2.15] | 1.40 [0.61, 3.23] | 1.20 [0.54, 2.70] | 1.88[0.57, 6.28] |
| *Working status* | | | | | | |
| Employed (Unemployed, OR = 1) | 0.61 [0.19, 1.91] | 1.77 [0.79, 4.00] | 1.83 [0.76, 4.39] | **0.27 [0.11, 0.68]** | 0.95 [0.40, 2.25] | **0.18[0.05, 0.68]** |
| Wages in Taka | | | | | | |
| 5000–10000 (< 5000, OR = 1) | 1.33[0.49, 3.65] | 0.98 [0.49, 1.99] | 0.98 [0.46, 2.08] | 1.89 [0.85, 4.18] | 0.93 [0.43, 1.98] | 1.58[0.47, 5.23] |
| 10000+ | 1.22 [0.39, 3.85] | 0.55 [0.23, 1.29] | 0.74 [0.30, 1.85] | **2.77 [1.06, 7.19]** | 0.94 [0.38, 2.33] | **4.30[1.06, 7.44]** |
| Living status | | | | | | |
| Without family (With family, OR = 1) | **2.71 [1.21, 6.09]** | 0.72 [0.40, 1.30] | 0.71 [0.37, 1.34] | 1.12 [0.58, 2.13] | 1.15 [0.60, 2.20] | 0.37 [0.13, 1.05] |
| Human rights | | | | | | |
| Economic right | | | | | | |
| Yes (No, OR = 1) | - | 1.30 [0.74, 2.28] | 1.01[0.55, 1.87] | 1.22 [0.66, 2.27] | 0.93 [0.51, 1.70] | 2.24 [0.93, 5.41] |
| Health right | | | | | | |
| Yes (No, OR = 1) | 0.95 [0.52, 1.75] | 0.62 [0.38, 1.01] | - | **4.44 [2.60, 7.57]** | **2.12 [1.27, 3.53]** | **0.38 [0.17, 0.86]** |
| Employment right | | | | | | |
| Yes (No, OR = 1) | 1.29 [0.73, 2.28] | - | 0.62 [0.38, 1.02] | 1.14 [0.69, 1.88] | 1.06 [0.65, 1.74] | **0.32 [0.15, 0.66]** |
| Right to education | | | | | | |
| Yes (No, OR = 1) | 1.34 [0.73, 2.47] | 1.10 [0.67, 1.80] | **4.59[2.67, 7.91]** | - | **2.43 [1.41, 4.18]** | 1.47 [0.66, 3.31] |
| Social and Civil right | | | | | | |
| Yes (No, OR = 1) | 1.01[0.55, 1.84] | 1.05 [0.64, 1.70] | **2.08 [1.25, 3.46]** | **2.41 [1.39, 4.16]** | - | **2.72 [1.15, 6.41]** |
| Political right | | | | | | |
| Yes (No, OR = 1) | 1.96[0.85, 4.50] | **0.41 [0.21, 0.79]** | **0.38 [0.18, 0.79]** | 1.80 [0.88, 3.71] | **2.45 [1.15, 5.22]** | - |

Bangladeshi Hijras [17] which found that well-educated and trained Hijras opted for voluntary with non-governmental organizations due to their inability to secure government jobs. In another study [28], educated Hijras were less likely to secure gainful employment in the formal sector, suggesting a lack of legal recourse for discrimination among gender identity, including the Hijras in Bangladesh.

The higher odds of violations to good health and the lack of access to adequate medical health reported among the Hijra community in this study may be linked to the lack of legal identification documents due to discrimination of the Hijra community [29]. Whilst a little over half of the respondents in this study identified as transgender, a substantial proportion identified as homosexuals and bisexuals. These findings confirm the complexity of perceived sexuality and self-identity by the Hijra community as opposed to the lay view that Hijras are characterized only by their feminine behavioral traits and the struggle to accept anything outside of the male-female gender dichotomy.

Hijras who did not live with their families reported higher odds of economic violations than those who lived with their families. It has been reported that unusual feminine behaviors

during early childhood by Hijras could bring shame to their families, making the family exclude them from family-related events [30]. In many cases, family members have made fun of these feminine behaviors and strongly oppose this behavior during adolescence [12, 30]. While Hijras generally dress as women, some may present themselves as men in different circumstances in order to avoid exploitation, harassment, and abandonment. Hijra members can, and some do, move between normative masculinity and Hijra [31]. For instance, the *Sadrali Hijra* in Bangladesh are non-emasculated and is often in a heterosexual marital relationship. The centrality of emasculation as a hallmark of being a Hijra is a remnant of the British colonization and may be based on the interpretation of the Quran. This became a barrier to how Bangladesh implemented its recent legislative change when Hijra applicants were terminated from the Ministry of Social Welfare for possessing male genitalia [4, 31]. Social exclusion and discrimination of the Hijras in Bangladesh should be discouraged at the family level. Family members, including parents, should have a positive attitude towards the Hijras from early childhood to after attaining puberty, including those born as hermaphrodites.

This study has several strengths and limitations. First, this is the first quantitative study that examined factors associated with the human rights status of the Hijra community after the Government of Bangladesh landmark decision of making Hijras the third gender in November 2013. Second, this study highlighted the gap in meeting the UN SDG 10 of reducing inequalities by 2030 in Bangladesh. Third, the study represents the wider view of the Bangladeshi Hijra population because it was conducted in the five large urban cities with the highest population of the Hijras in Bangladesh. Lastly, the six human rights items showed good internal consistency, and thus, these items could be used for future studies in this community. However, this study also has some limitations. First, the cross-sectional study design limits causal inference and recall bias may have influenced our findings due to self-reported data. Second, some misclassification bias may have occurred, leading to an overestimation or underestimation of the associated factors. Third, we did not account for some of the confounding factors, such as participation in HIV prevention programs, family relationships, community connectedness, which may have affected the factors associated with the status of human rights of the Hijra community in Bangladesh. Lastly, Hijras in less developed cities, including those living in rural and remote Bangladesh, were excluded from the current survey, and the results may not reflect their opinion.

## Conclusions

This study's findings suggest considerable variations in the prevalence of the six human rights violations of the Hijras, with the highest prevalence for economic rights violations of the Hijras and the lowest prevalence for social and civil rights violations. The study indicated that Hijras, who earned more than 10,000 Bangladesh Taka and those living with family members, had a higher likelihood of reporting human rights violations in Bangladesh. The Hijra community is diverse, and its members share commonalities as a function of their status as sexual minorities and concomitant issues of marginalization, discrimination, and heterosexism. This study suggests the need to implement interventional programs at school, community, and national levels. Policy interventions can be introduced at the school level to support sexually, and gender diverged students and awareness campaigns organized within school communities to tackle anti-Hijra harassment within the school environment. At the community level, early interventions such as media campaigns on awareness and acceptance campaigns are needed to inform the public on the challenges faced by members of the Hijra communities. In contrast, increased access to education, greater representation of Hijras in government offices are needed at the national level. These interventions are necessary if the Government is working

towards improving the social inclusion of Hijras to meet the UN SDG 10 of reducing inequalities by 2030 in Bangladesh.

## Supporting information

**S1 Data.**
(CSV)

## Author Contributions

**Conceptualization:** A. S. M. Amanullah.

**Data curation:** A. S. M. Amanullah, Tanvir Abir, Uchechukwu L. Osuagwu, Giasuddin Ahmed, Kingsley E. Agho.

**Formal analysis:** A. S. M. Amanullah, Tanvir Abir, Taha Husain, Uchechukwu L. Osuagwu, Giasuddin Ahmed, Kingsley E. Agho.

**Investigation:** Giasuddin Ahmed.

**Methodology:** Tanvir Abir, Uchechukwu L. Osuagwu, Kingsley E. Agho.

**Project administration:** A. S. M. Amanullah, Dewan Muhammad Nur -A Yazdani.

**Visualization:** Saleh Ahmed, Dewan Muhammad Nur -A Yazdani.

**Writing – original draft:** A. S. M. Amanullah, Taha Husain, David Lim, Dewan Muhammad Nur -A Yazdani, Kingsley E. Agho.

**Writing – review & editing:** David Lim, Saleh Ahmed, Dewan Muhammad Nur -A Yazdani, Kingsley E. Agho.

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
