## [Decision Letter · Decision Letter 0]

20 Jul 2021

PONE-D-21-17167

Human Rights Violations and Associated Factors of the Hijras in Bangladesh – A Cross-sectional study

PLOS ONE

Dear Dr. ABIR,

Thank you for submitting your manuscript to PLOS ONE. After careful consideration, we feel that it has merit but does not fully meet PLOS ONE’s publication criteria as it currently stands. Therefore, we invite you to submit a revised version of the manuscript that addresses the points raised during the review process.

We look forward to receiving your revised manuscript.

Kind regards,

Enamul Kabir

Academic Editor

PLOS ONE

Journal Requirements:

"NO: The authors did not receive support from any organization for the submitted work."

We note that one or more of the authors are employed by a commercial company: "Legacy Global Consultancy,"

5. Please upload a copy of Figure 1, to which you refer in your text on line 208. If the figure is no longer to be included as part of the submission please remove all reference to it within the text.

Reviewers' comments:

Reviewer's Responses to Questions

**Comments to the Author**

1. Is the manuscript technically sound, and do the data support the conclusions?

Reviewer #1: Yes

Reviewer #2: Partly

Reviewer #3: Yes

Reviewer #4: Yes

2. Has the statistical analysis been performed appropriately and rigorously? 

Reviewer #1: Yes

Reviewer #2: No

Reviewer #3: Yes

Reviewer #4: Yes

3. Have the authors made all data underlying the findings in their manuscript fully available?

Reviewer #1: Yes

Reviewer #2: Yes

Reviewer #3: Yes

Reviewer #4: Yes

4. Is the manuscript presented in an intelligible fashion and written in standard English?

Reviewer #1: Yes

Reviewer #2: Yes

Reviewer #3: No

Reviewer #4: Yes

5. Review Comments to the Author

Reviewer #1: In Introduction Authors have given good account of history about Hijras dating back to Mahabharatha and to the present. They have explained about sexual, social and economic difficulties faced by Hijras

The study design and the formula used are appropriate for the study

Univariate analysis of factors associated with the six human right violations explained well.

Hijras staying without their families suffering more - is a fact and well recorded

The positive aspect of this study is the authors have explained the strengths and limitations of their study

Suggestion: In their future study they may conduct similar study with normal males females as their study subjects so as to know their behavioral aspects towards Hijras and their opinion about Hijras rights in the society

Reviewer #2: Introduction: Needs improvements in its structure. I believe that the entire historical process is important, however, in the end, I cannot have a dimension of the research problem. It is necessary to clarify the search problem in the introduction.

Methods: Regarding the sample, I believe it needs more detail in the selection process.

Needs improvement in the outcome variable. How was it created? This is not clear.

Statistical analyzes need more detail. How did you carry out the process of creating the models?

Did you check the final fit of the model? Hosmer-Lemeshow goodness-of-fit method?

What is the percentage of explanation of the variance of each adjusted model?

Results: When Cronbach's alpha was less than 0.7%, what was done with the variable? Needs detailing.

Figure 1 did not come in the file.

Discussion:

Needs more detail of the findings compared to the literature.

The limitation of the study design is inherent to the design. It does not constitute a limitation of your study.

Reviewer #3: The topic is important. However, the reviewer has some observations:

General: Writing could be improved. Examples are numerous.

The first sentence in abstract- '..........to preserve the rights of their right' is not clear. Needs to be rephrased.

In the Introduction section, the authors should briefly rationalize why they have measured 'the six human rights violations' among the Hijras in Bangladesh, with appropriate reference(s).

Please split the paragraph under the heading 'Setting'. It is suggested to keep all the texts, from the sentence 'prior to data collection........', under a different heading 'Data collection'.

It is not clearly mentioned how the samples were selected from each district. Equally from each district? (line 153)

Please briefly write about Bandhu Social Welfare Society (line 154, page 8).

Please keep consistency in mentioning the name of the organization 'Bandhu Social Welfare Society' throughout the paper. You have written only 'Bandhu' at page 8 line 156. Better to use acronym (if any).

Please mention the total number of Hijras in the sampling frame provided by the organization. Example, out of XXX, you have randomly selected 346 (line 158).

The paragraph written under the 'Study design' is more about data collection tool/data collection, not about study design. The study is simply a 'cross-sectional' in design. Please move the paragraph under a different heading 'Data collection/data collection tool'.

It is suggested to write covariates, rather than confounding variables. (line 175).

Figure 1 is missing! I don't see Figure 1 in the document.

How is 73% 'about one-third'(line 209). Please make correction.

Different human rights reported under 'Table 2' are not participant characteristics. Please remove the estimates of human rights violations from 'Table 2' and report those under a different Table.

One of the objectives of this study is to identify the factors associated with primary outcomes (violation of human rights). So, this is not clear why the authors have shown the association of one primary outcome with other primary outcomes in univariate and multivariate regression model. Example: Economic right with health rights.

Reviewer #4: General Comments

It is a well-written article, and It is relevant to the area. E trata de um tema de suma importância e que merece ser amplamente debatido pela comunidade científica mundial.

However, we found some aspects that deserve to be better clarified

Comments

1. Introduction

• It is a well-written introduction, but too long. Try to cut it down a bit and transfer some parts to the discussion.

• Page 13 - Lines 139-141 – Check whether it would not be more accurate to report that the assessment of the violation of the Hijras' human rights was by self-report.

2. Methods

• Page 14 - Lines 152-153 – Explain why these five districts of Bangladesh were selected, and give more characteristics of these regions;

• Make the instrument available for analysis through some link.

3. Results

- It is always important to emphasize that the human rights violations were self-reported by the participants. To make clear how the events were verified.

- Was any assessment made of the severity of the human rights violations in each of the dimensions analyzed? I believe that there may have been violations of different severity

• Page 16 - Lines 204 – Convert the income to US$ (both in the text and in the Table), and state whether it is below or above the average income of workers in Bangladesh.

• Page 18 – Line 239 – Apparently there is an abrupt change in the ODDs for political rights violations in those who earned more than 10000 takas, when it came to multivariate analysis. Very strange this result. What could explain it?

• Page 23 – table 2 – Why didn't you subdivide the age groups into more categories? What is the reason for using only 15-25 and 26+?

6. PLOS authors have the option to publish the peer review history of their article (what does this mean?). If published, this will include your full peer review and any attached files.

Reviewer #1: No

Reviewer #2: No

Reviewer #3: No

Reviewer #4: No

---

## [Author Response · Author response to Decision Letter 0]

22 Jan 2022

The topic is essential. However, the reviewer has some observations:

1. General: Writing could be improved. Examples are numerous. 

Response: The writing of the revised manuscript has been improved. See the highlighted changes across the manuscript.

2. The first sentence in the abstract- '..........to preserve the rights of their right' is not clear. It needs to be rephrased.

Response: Done. The Section was revised.

3. In the Introduction section, the authors should briefly rationalize why they have measured 'the six human rights violations' among the Hijras in Bangladesh, with appropriate reference(s).

Response: The following changes have been made;

The authors selected human rights components by following the significant articles of The Universal Declaration of Human Rights (UDHR) proclaimed by the United Nations General Assembly in Paris on 10 December 1948 (General Assembly resolution 217 A). Civil, political, economic, cultural, health, and social rights are the legal bindings for the United Nations member states [25]. 

4. Please split the paragraph under the heading 'Setting'. It is suggested to keep all the texts, from the sentence 'before data collection........', under a different heading 'Data collection'.

Response: Done. The data collection subheading was created and repositioned in the manuscript.

5. It is not mentioned how the samples were selected from each district. Equally from each district? (line 153)

Response: The sentence below was included in the text: 

"To be representative, the equal number of participants were selected from each district".

6. Please briefly write about Bandhu Social Welfare Society (line 154, page 8).

Response: We have provided some information about the Bandhu Social Welfare Society.

7. Please keep consistency in mentioning the name of the organization 'Bandhu Social Welfare Society' throughout the paper. You have written only 'Bandhu' at page 8 line 156. Better to use acronym (if any).

Response: We have maintained Bandhu Social Welfare Society across the manuscript.

8. Please mention the total number of Hijras in the sampling frame provided by the organization. Example, out of XXX, you have randomly selected 346 (line 158).

Response: The organization provided us with a list of 1600 Hijras. From the list, 346 Hijras were selected randomly. 

9. The paragraph written under the 'Study design' is more about data collection tool/data collection, not about study design. The study is simply a 'cross-sectional' in design. Please move the paragraph under a different heading 'Data collection/data collection tool'.

Response: The Section has been revised, and the Section on 'study design' section moved into 'data collection.'

10. It is suggested to write covariates, rather than confounding variables. (line 175).

Response: Done

11. Figure 1 is missing! I don't see Figure 1 in the document.

Response: Figure 1 was included as a separate document in the previous submission. It is now inserted in the main manuscript text for quick visualization.

12. How is 73% 'about one-third'(line 209). Please make correction.

Response: This was a typo error and has been corrected.

13. Different human rights reported under 'Table 2' are not participant characteristics. Please remove the estimates of human rights violations from 'Table 2' and report those under a different Table.

Response: Table 2 showed the variables used in regression analysis. We have renamed Table 2 to reflect the contents. The new title reads: 

a. Participant Characteristics and the categories of human rights violations (n=346).

No further change was made in the table

14. One of the objectives of this study is to identify the factors associated with primary outcomes (violation of human rights). So, this is not clear why the authors have shown the association of one primary outcome with other primary outcomes in univariate and multivariate regression model. Example: Economic right with health rights.

Response: Agreed and for clarity, now reads "six human rights violation indicators."

---

## [Decision Letter · Decision Letter 1]

14 Mar 2022

PONE-D-21-17167R1Human Rights Violations and Associated Factors of the Hijras in Bangladesh – A Cross-sectional studyPLOS ONE

Dear Dr. ABIR,

Thank you for submitting your manuscript to PLOS ONE. After careful consideration, we feel that it has merit but does not fully meet PLOS ONE’s publication criteria as it currently stands. Therefore, we invite you to submit a revised version of the manuscript that addresses the points raised during the review process.

We look forward to receiving your revised manuscript.

Kind regards,

Enamul Kabir

Academic Editor

PLOS ONE

Journal Requirements:

Reviewers' comments:

Reviewer's Responses to Questions

**Comments to the Author**

1. If the authors have adequately addressed your comments raised in a previous round of review and you feel that this manuscript is now acceptable for publication, you may indicate that here to bypass the “Comments to the Author” section, enter your conflict of interest statement in the “Confidential to Editor” section, and submit your "Accept" recommendation.

Reviewer #1: All comments have been addressed

Reviewer #2: (No Response)

Reviewer #3: All comments have been addressed

2. Is the manuscript technically sound, and do the data support the conclusions?

Reviewer #1: Yes

Reviewer #2: Yes

Reviewer #3: Yes

3. Has the statistical analysis been performed appropriately and rigorously? 

Reviewer #1: Yes

Reviewer #2: No

Reviewer #3: Yes

4. Have the authors made all data underlying the findings in their manuscript fully available?

Reviewer #1: Yes

Reviewer #2: Yes

Reviewer #3: Yes

5. Is the manuscript presented in an intelligible fashion and written in standard English?

Reviewer #1: Yes

Reviewer #2: Yes

Reviewer #3: Yes

6. Review Comments to the Author

Reviewer #1: The Authors have revised the manuscript as per the instructions of the reviewers. They have added/gave explanations wherever necessary in the text. No more corrections

Reviewer #2: Methods: Regarding the sample, I believe it needs more detail in the selection process.

Needs improvement in the outcome variable. How was it created? This is not clear.

Statistical analyzes need more detail. How did you carry out the process of creating the models?

Did you check the final fit of the model? Hosmer-Lemeshow goodness-of-fit method?

What is the percentage of explanation of the variance of each adjusted model?

Reviewer #3: (No Response)

7. PLOS authors have the option to publish the peer review history of their article (what does this mean?). If published, this will include your full peer review and any attached files.

Reviewer #1: No

Reviewer #2: No

Reviewer #3: No

---

## [Author Response · Author response to Decision Letter 1]

22 Mar 2022

Response to reviewer’s comments

We thank the reviewer for the very useful comments. We have addressed the comments below and used red fonts to highlight the sections in the manuscript where the requested changes were made.

Reviewer #2: 

Question: Methods: Regarding the sample, I believe it needs more detail in the selection process.

Response: we have added the district sampled and indicated the reason for using PPS sampling method in our study.

Question: Needs improvement in the outcome variable. How was it created? This is not clear.

Response: We have added more to this section of the paper for clarity. The revised section now reads:

Outcome variables

The six human rights violations of economic, employment, health, education, political, social/civil rights were the outcome variables in this study. There were three items on the economic right violations, twelve on right to employment, six on right to health, five items each on violations to education and political rights, and eleven on social/civil rights of the participants (see Table 1 for details). For each of the items, a positive response was coded as ‘1’ otherwise ‘0’ for a negative response. A human right violation was recorded if the participant had responded positively to all of the items in that section (i.e. a coding of ‘1’ to all items in that section). For example, on economic right, a violation was recorded only when ‘the items Eco-1, Eco-2 and Eco-2 in Table 1 were each coded as’1’, whereas any coding of ‘0’ in Eco-1, Eco-2 or Eco-2 was considered no violation to the economic right of the respondent. Similar coding method was applied to the employment, health, education, political, social/civil right violations. 

Question: Statistical analyses need more detail. How did you carry out the process of creating the models?

Response: We used “collin” command in STATA to check for collinearity and all the variables VIF were less than 4 which indicates that there is no collinearity. We also included all the covariates in the multiple logistic regression analysis at once. We conducted another analytical method by first doing a univariable logistic regression analysis including all the covariates and those with P value < 0.20 were retained and were used to build a multiple logistic regression model. For the multiple logistic regression, a manual backward elimination procedure was applied to remove non-significant variables (P > 0.05). The factors associated with each outcome variables obtained using the two statistical methods were similar with the method reported in the manuscript producing more associated factors. See Table 1 below for when we used the suggested method. 

For the reason stated above, no change was made in the revised manuscript.

Table 1 – adjusted odds ratios - multiple logistic regression with retaining those with P value < 0.20 

Variables Economic rights Variables Social and Civil rights

Demography AOR[95%CI] Health right AOR[95%CI]

Dhaka/ Gazipur (Non-Dhaka^, OR=1) 0.11 [0.06, 0.20] Yes (No, OR=1) 1.78 [1.10, 2.87]

Hijra Communities - Right to education 

bisexuality (homosexuality, OR=1) 2.66 [1.29, 5.46] Yes (No, OR=1) 2.92 [1.73, 4.92]

Educational status - Variables Political rights

Secondary (Primary, OR=1) 0.56 [0.26, 1.20] District AOR[95%CI]

Tertiary 0.26 [0.10, 0.63] Dhaka/ Gazipur (Non-Dhaka^, OR=1) 0.01 [0.00, 0.01]

Living status - Hijra Communities 

Without family (With family, OR=1) 2.17 [1.02, 4.58] bisexuality (homosexuality, OR=1) 0.02[0.01, 0.06]

Variables Health rights Working status 

Right to education AOR[95%CI] Employed (Unemployed, OR=1) 0.11[0.03, 0.36]

Yes (No, OR=1) 3.76[2.28, 6.19] Wages in Taka 

Social and Civil right 5000-10000 (< 5000, OR=1) 1.97 [0.61, 6.31]

Yes (No, OR=1) 1.78[1.09, 2.92] 10000+ 5.68 [1.47, 22.01]

Political right Health right 

Yes (No, OR=1) 0.54 [0.34, 0.87] Yes (No, OR=1) 0.48 [0.23, 0.99]

Variables Education rights Employment right 

Employed (Unemployed, OR=1) AOR[95%CI] Yes (No, OR=1) 0.33 [0.17, 0.68]

Wages in Taka 0.28 [0.13, 0.63] Social and Civil right 

5000-10000 (< 5000, OR=1) 1.93 [0.89, 4.20] Yes (No, OR=1) 2.58 [1.17, 5.67]

10000+ 3.11 [1.32, 7.36] Variables Employment rights

Health right Political right AOR[95%CI]

Yes (No, OR=1) 3.77 [2.29, 6.22] Yes (No, OR=1) 0.57 [0.37, 0.88]

Social and Civil right 

Yes (No, OR=1) 2.69 [1.57, 4.60] 

Question: Did you check the final fit of the model? Hosmer-Lemeshow goodness-of-fit method?

Response: Hosmer-Lemeshow goodness-of-fit was not used in this study because it is considered to be obsolete by statisticians for the following reasons: (1) It does not properly take overfitting into account, (2) poor power in small data sets and, (3) dependence on arbitrary binning of predicted probabilities (i.e. the value 10 is arbitrary) and often has power that is too low (Fagerland & Hosmer, 2012; Hosmer, 1997). However and for your record, we have produced the Hosmer-Lemeshow goodness-of-fit for the 6 outcome variables and the result indicated a good fit (P > 0.05) for all the six human right violations outcomes examined in this study (see Table 2 below). Hence, no change was made to the manuscript base on the limitations discussed above, unless the editor thinks it is necessary as a supplementary table while noting the limitations.

Table 2 Hosmer-Lemeshow goodness-of-fit for the 6 human right outcomes

Human right variables chi-square P-value

Economic 7.5 0.481

Employment 11.0 0.201

Health 4.8 0.783

Education 1.6 0.980

Social and Civil 13.4 0.100

Political 10.5 0.231

References:

Fagerland, M. W., & Hosmer, D. W. (2012). A generalized Hosmer–Lemeshow goodness-of-fit test for multinomial logistic regression models. The Stata Journal, 12(3), 447-453.

Hosmer, D.W. (1997). "A comparison of goodness-of-fit tests for the logistic regression model". Stat Med. 16 (9): 965–98

Question: What is the percentage of explanation of the variance of each adjusted model?

Response: We did not report variance for each of our adjusted model because: (1) Logistic regression will always be heteroscedastic – the error variances differ for each value of the predicted score. (2) For each value of the predicted score there would be a different value of the proportionate reduction in error. Therefore, it is inappropriate to think of R² as a proportionate reduction in error in logistic regression (Cohen et al. 2002). However, and for your record, we have produced the R² for the 6 human right outcomes (see Table 3 below). Hence, no change was made to the manuscript base on the limitations discussed above.

Table 3 Maximum Likelihood and Pseudo R² for the 6 human right outcomes

Human right variables Maximum Likelihood R2 Pseudo R2

Economic 0.200 0.194

Employment 0.060 0.045

Health 0.185 0.150

Education 0.191 0.157

Social and Civil 0.115 0.095

Political 0.517 0.538

Reference:

Cohen, Jacob; Cohen, Patricia; West, Steven G.; Aiken, Leona S. (2002). Applied Multiple Regression/Correlation Analysis for the Behavioral Sciences (3rd ed.)

---

## [Decision Letter · Decision Letter 2]

20 May 2022

Human Rights Violations and Associated Factors of the Hijras in Bangladesh – A Cross-sectional study

PONE-D-21-17167R2

Dear Dr. ABIR,

We’re pleased to inform you that your manuscript has been judged scientifically suitable for publication and will be formally accepted for publication once it meets all outstanding technical requirements.

Kind regards,

Enamul Kabir

Academic Editor

PLOS ONE

Additional Editor Comments (optional):

Reviewers' comments:

Reviewer's Responses to Questions

**Comments to the Author**

1. If the authors have adequately addressed your comments raised in a previous round of review and you feel that this manuscript is now acceptable for publication, you may indicate that here to bypass the “Comments to the Author” section, enter your conflict of interest statement in the “Confidential to Editor” section, and submit your "Accept" recommendation.

Reviewer #1: All comments have been addressed

Reviewer #3: (No Response)

2. Is the manuscript technically sound, and do the data support the conclusions?

Reviewer #1: Yes

Reviewer #3: Yes

3. Has the statistical analysis been performed appropriately and rigorously? 

Reviewer #1: Yes

Reviewer #3: Yes

4. Have the authors made all data underlying the findings in their manuscript fully available?

Reviewer #1: Yes

Reviewer #3: Yes

5. Is the manuscript presented in an intelligible fashion and written in standard English?

Reviewer #1: Yes

Reviewer #3: No

6. Review Comments to the Author

Reviewer #1: The Authors have answered the queries raised by the reviewers and incorporated the required data and details in the text. No more corrections

Reviewer #3: (No Response)

7. PLOS authors have the option to publish the peer review history of their article (what does this mean?). If published, this will include your full peer review and any attached files.

Reviewer #1: No

Reviewer #3: No

---

## [Editor Report · Acceptance letter]

13 Jun 2022

PONE-D-21-17167R2 

Human Rights Violations and Associated Factors of the Hijras in Bangladesh – A cross-sectional study 

Dear Dr. Abir:

I'm pleased to inform you that your manuscript has been deemed suitable for publication in PLOS ONE. Congratulations! Your manuscript is now with our production department. 

Kind regards, 

on behalf of

Dr. Enamul Kabir 

Academic Editor

PLOS ONE